

# Mass spectrometric measurements of ambient ions and estimation of gaseous sulfuric acid in the free troposphere and lowermost stratosphere during the CAFE-EU/BLUESKY campaign

Marcel Zauner-Wieczorek[1], Martin Heinritzi[1], Manuel Granzin[1], Timo Keber[1], Andreas Kürten[1], Katharina Kaiser[2], Johannes Schneider[2], and Joachim Curtius[1]

[1]Institute for Atmospheric and Environmental Sciences, Goethe University Frankfurt am Main, Frankfurt am Main, 60629, Germany
[2]Particle Chemistry Department, Max Planck Institute for Chemistry, Mainz, 55128, Germany

*Correspondence to*: Marcel Zauner-Wieczorek (zauner-wieczorek@iau.uni-frankfurt.de)



**Abstract.** Ambient ions play an important role in atmospheric processes such as ion-induced new particle formation. While there are several studies of ambient ions for different layers of the atmosphere, data coverage for the free troposphere and especially the upper troposphere and lower stratosphere (UTLS) region is scarce. Here, we present the first airborne measurements of ambient ions using a High Resolution-Atmospheric Pressure interface-Time Of Flight-Mass Spectrometer (HR-APi-TOF-MS) in the free troposphere and lower stratosphere above Europe on board the HALO aircraft during the CAFE-EU/BLUESKY campaign in May and June 2020. In negative measurement mode, we observed nitrate and hydrogen sulfate and their related ion clusters in an altitude range of 4.7 to 13.4 km. The horizontal profiles for those ions reveal an increasing count rate for $NO_3^-$ and $(HNO_3)NO_3^-$ towards higher altitudes, but no significant trend for $HSO4^-$. From the count rates of the nitrate ($NO_3^-$) and hydrogen sulfate ($HSO_4^-$) core ions, we inferred the number concentration of gaseous sulfuric acid. The lowest average value was found to be $1.8 \cdot 10^5$ $cm^{-3}$ at the maximum altitude bin, i.e. 13.4 km. The highest average value of $9.1 \cdot 10^5$ $cm^{-3}$ was observed in the 8.7–9.2 km altitude bin. During the transit through a liquid water cloud, we observed an event of enhanced ion count rates and aerosol particle concentrations that can largely be assigned to nitrate ions and particles, respectively; this may have been caused by the shattering of cloud droplets on the surface of the aircraft or the inlet. Furthermore, we report the proof of principle for the measurement of ambient cations and the identification of protonated pyridine.

## 1 Introduction

Earth's atmosphere can be considered a diluted plasma, as it not only contains neutral gases, but also ions in the gas phase. These ions play a crucial role in several atmospheric processes such as ion-induced nucleation of aerosol particles (Hirsikko et al., 2011). For a long time, the measurement of atmospheric ions has been the only means to infer qualitative and quantitative information on the composition of certain neutral compounds of the atmosphere; as an example, the atmospheric concentration of sulfuric acid, one of the most important substances involved in new particle formation (Seinfeld and Pandis, 2006), has historically been determined using ambient ion data.

The most important sources of ionisation from ground level up to about 50 km altitude are galactic cosmic rays (GCRs), and, close to the Earth's surface, the radioactive decay of radon (Viggiano and Arnold, 1995; Bazilevskaya et al., 2008). GCRs consist mostly of protons and α particles and these are able to penetrate the Earth's atmosphere. When their energy is high enough, secondary pions and muons are formed in nuclear-electromagnetic cascades. These secondary pions and muons interact with atmospheric compounds and cause their ionisation (Bazilevskaya et al., 2008). Predominantly, $N_2^+$, $O_2^+$, $N^+$, and $O^+$ cations are formed along with electrons. The electrons interact rapidly with oxygen, forming $O_2^-$ and $O^-$ (Arijs, 1992; Arnold and Knop, 1987; Shuman et al., 2015; Viggiano and Arnold, 1995). The cations quickly react to $NO^+$ and, subsequently, to proton hydrates, i.e. $H^+(H_2O)_n$. The anions react to $(H_2O)_nCO_3^-$ and, subsequently, to $(H_2O)_nNO_3^-$ (Arijs, 1992; Shuman et al., 2015). As the aforementioned reactions occur so rapidly and the lifetimes of the resulting ion clusters, i.e. $H^+(H_2O)_n$ and $(H_2O)_nNO_3^-$, are relatively long, they can be considered the starting points for the ion chemistry of the stratosphere and



troposphere (Shuman et al., 2015; Viggiano and Arnold, 1995). For further details of the ionisation in the atmosphere, the article by Bazilevskaya et al. (2008) provides an overview, whilst, for the ion chemistry of the upper troposphere, Shuman et

al. (2015) offer a comprehensive review.

The subsequent negative ion chemistry in the troposphere and stratosphere is comparably simple. The water-nitrate clusters, $(H_2O)_nNO_3^-$, are able to take up nitric acid or exchange water molecules with nitric acid, yielding clusters of the type $(HNO_3)_m(H_2O)_nNO_3^-$ (Shuman et al., 2015; Viggiano and Arnold, 1995). Acids, HX, with a larger gas-phase acidity than $HNO_3$ can participate in a charge transfer reaction with $NO_3^-$ and ligand exchange reactions with $HNO_3$, leading to clusters of

the type $(HX)_k(HNO_3)_m(H_2O)_nX^-$. For most of the troposphere and stratosphere, only sulfuric acid $(H_2SO_4)$ fulfils this requirement while being sufficiently abundant (Viggiano and Arnold, 1995). It is only closer to the ground that strong and abundant acids such as malonic acid $(CH_2(COOH)_2)$, methanesulfonic acid $(CH_3SO_3H)$, or iodic acid $(HIO_3)$ can also be observed as part of the ion clusters (Ehn et al., 2010; Eisele, 1989a; Frege et al., 2017; Viggiano and Arnold, 1995; He et al., 2021). It should be noted that, practically, the only source for $HSO_4^-$ ions and their ion clusters is the reaction of gaseous

sulfuric acid with nitrate (clusters) (Viggiano and Arnold, 1995). The direct ionisation of sulfuric acid is negligible. Throughout the troposphere and stratosphere, ion clusters with an $NO_3^-$ core ion are predominant, except for the altitude range of around 35 to 40 km; here, the $HSO_4^-$ core ion family is predominant, however, they are also abundant throughout the whole troposphere and stratosphere, but to a lesser extent than the $NO_3^-$ core ion family (Viggiano and Arnold, 1995).

The knowledge of the abundance of the neutral species $HNO_3$ and $H_2SO_4$ in the upper parts of the atmosphere is mainly derived

from calculations based on the measurements of the aforementioned ambient ions (Arnold and Fabian, 1980; Heitmann and Arnold, 1983; Arnold and Qiu, 1984). This calculation will be discussed in detail later. Direct measurements of the neutral trace compounds have been few and their results are only available for a limited part of the atmosphere. Measurements of neutral gaseous sulfuric acid above the boundary layer have been performed at various altitude ranges: 3 to 11 km above the Arctic (Möhler and Arnold, 1992), 0 to 7.5 km in mid- and polar latitudes (Mauldin et al., 2003), 0 to 6 km above the southern

Pacific Ocean (Weber et al., 1999; Weber et al., 2001), 0 to 7 km above the western Pacific Ocean (Weber et al., 2003), and on two mountain research sites in the Alps (Zugspitze, Germany, 2650 m a.s.l., Aufmhoff et al. (2011); Jungfraujoch, Switzerland, 3580 m a.s.l., Bianchi et al. (2016)). The reported concentrations of sulfuric acid are high in the boundary layer ($> 10^7$ cm$^{-3}$ possible), low in the free troposphere ($4 \cdot 10^5$ to $4 \cdot 10^6$ cm$^{-3}$) and slightly enhanced in the tropopause; however, in the lower stratosphere, the concentrations are lower again ($10^5$ to $10^6$ cm$^{-3}$), according to Möhler and Arnold (1992). The

most recent airborne measurements of neutral compounds using a chemical ionisation source in the negative ion mode have been performed by our group during the CAFE-EU/BLUESKY campaign in 2020; the results will be presented in detail elsewhere. Nitric acid measurements in the upper troposphere and lower stratosphere (UTLS) showed that $HNO_3$ concentrations are low in the free troposphere and enhanced in the stratosphere (Schneider et al., 1998; Schumann et al., 2000; Jurkat et al., 2016).

Positive ion chemistry in the troposphere and stratosphere, on the other hand, involves acetonitrile, pyridine, ammonia and amines (Viggiano and Arnold, 1995). Bases, B, with higher proton affinities than water, if sufficiently abundant, can react with





the proton hydrates via ligand exchange to form clusters of the type $H^+(H_2O)_n(B)_m$. One prominent B compound found throughout the troposphere and stratosphere is acetonitrile ($CH_3CN$), while around the tropopause, methanol ($CH_3OH$) and acetone (($CH_3)_2CO$) also become important, and, in the lower troposphere, ammonia ($NH_3$), pyridine ($C_5H_5N$) and several

amines are molecules which are found to form clusters of the aforementioned type (Viggiano and Arnold, 1995). Ascending from ground level towards higher altitudes, the predominant positive core ions are ammonium, protonated pyridine and protonated amines (0–5 km), protonated acetone (5–12 km), protonated acetonitrile (12–30 km), and proton hydrates (> 30 km) (Viggiano and Arnold, 1995).

The first rocket-borne mass spectrometric measurements of atmospheric ions were conducted in the mesosphere and

ionosphere (above 64 km) (Johnson et al., 1958; Narcisi and Bailey, 1965; Narcisi et al., 1971; Arnold et al., 1971), leading to the identification of key species such as $O^+$, $O_2^+$, $H_3O^+$, $O_2^-$, $CO_3^-$, and $NO_3^-$, amongst others. In subsequent years, the investigations were extended to lower parts of the atmosphere and included an ever-growing number of identified ions. Arnold et al. (1977) reported the first rocket-borne measurements in the stratosphere which were then followed by balloon-borne investigations (Arijs et al., 1978; Arnold et al., 1978; Arnold and Henschen, 1978). By employing mass spectrometers on

aircraft, the lower stratosphere and upper troposphere could also be sampled (Heitmann and Arnold, 1983; Arnold et al., 1984). In addition, in the 1980s, Fred Eisele's group performed ground-based experiments in the boundary layer and in the lower free troposphere on mountain research stations (Perkins and Eisele, 1984; Eisele, 1986, 1988, 1989a, 1989b). With all of these studies, measurements of the negative, as well as the positive, ions in the atmospheric range of 0 to more than 100 km altitude have been covered, unravelling a multitude of atmospheric compounds. Nevertheless, these measurements have been few and

questions of reproducibility and variability of the results cannot be fully resolved. In the beginning of the 1990s, the development of chemical ionisation sources allowed for the direct measurement of neutral trace gases of interest. Indirect measurements utilising the detection of atmospheric ions became scarcer. Since then, to our knowledge, only Möhler et al. (1993) have presented measurements of ambient ions in the UTLS (although these were combined with chemical ionisation measurements), while Eichkorn et al. (2002) observed the existence of large positive ion clusters with masses of up to 2500 Da

in the upper troposphere, providing evidence for the ion-induced nucleation of aerosol particles. In addition, Arnold et al. (1998) measured the gaseous ion composition in the exhaust plume of a jet aircraft in flight and derived the number concentration of gaseous sulfuric acid.

The introduction of high-resolution time-of-flight mass spectrometers (HR-TOF-MS) allowed for a better signal identification of mass spectra, which was also applied to the field of ambient ion measurements. Junninen et al. (2010) presented the first

HR-TOF-MS to measure the composition of atmospheric ions. Consequently, this group was also able to identify an enormous number of ambient ions via ground-based measurements in a boreal forest in Hyytiälä, Finland, including organic ion clusters (Ehn et al., 2010). Frege et al. (2017) characterised ambient ions in the lower part of the free troposphere with measurements at the Jungfraujoch research station in Switzerland at 3454 m a.s.l. The dominant negative ions identified were assigned to sulfuric, nitric, malonic and methanesulfonic acid. Positive ions were assigned to amines, ammonia and organic clusters. More

recently, Beck et al. (2021b) conducted airborne measurements of ambient ions above the boreal forest at an altitude range of



0 to 3200 m around the Station for Measuring Ecosystem Atmospheric Relations (SMEAR) II in Hyytiälä. They identified ions belonging to nitric, iodic, methanesulfonic, sulfuric, and carboxylic acid, as well as highly oxygenated organic molecules (HOMs). Furthermore, they reported diurnal variations of the ions depending on the changing atmospheric layers at different periods of the day. Beck et al. (2021a) also derived concentrations of gaseous sulfuric acid from the measurements of ambient
ions. They were able to validate their calculations with chemical ionisation mass spectrometric measurements for daytime in the boreal forest.

In this work, we present the first aircraft-borne measurements of ambient ions with an HR-TOF-MS in the UTLS during the CAFE-EU/BLUESKY campaign in May and June 2020, allowing for an unambiguous identification of the observed ions. We present the observed mass spectrum of negative ambient ions and show the altitude-dependence of the most important ions.
From the ion signals of nitrate and hydrogen sulfate, we derive an estimate for the gaseous sulfuric acid concentration. For the measurement of positive ambient ions, we provide a proof of principle and a first identification of observed signals.

## 2 Methods

### 2.1 Instruments

We used an HR-TOF-MS (Tofwerk AG, Thun, Switzerland) in combination with a new in-house developed and manufactured
Chemical Ionisation (CI) source, named Switchable CORona Powered ION source (SCORPION); this will be described in more detail in an upcoming publication (Heinritzi et al., in preparation). The main purpose of this setup was to enable nitrate reagent ion-based CI-TOF-MS measurements on board the High Altitude and LOng range research aircraft (HALO). However, by setting all voltages (including the corona voltage) in the SCORPION source to zero, i.e. to the same potential as the entrance pinhole of the TOF-MS, the measurement of ambient ions was enabled. This measurement mode is called the Atmospheric
Pressure interface (APi) mode. We used an inlet system that is almost completely manufactured from metal in order to avoid the build-up of electrostatic charge spots, which would then deflect the incoming ambient ions to the walls. The only non-metal piece in the inlet system was a 2 mm thick polyether ether ketone (PEEK) spacer that electrically isolates the inner parts of the ion source from the inlet tube. Our inlet system was designed to minimise wall losses; this is beneficial for both chemical ionisation and the ambient ion mode. We used the LiF-OH inlet system that was specifically developed for HALO (this is
usually used for OH measurements with the Laser-induced Fluorescence technique (LiF) (Broch, 2011)), as well as a 25 mm outer diameter, 1.7 m long stainless steel inlet tube with high flow (10 to 30 slpm, depending on the flight altitude) to connect the inlet with our instrument. The TOF-MS records data at a 1 Hz acquisition frequency in an $m/z$ range from 4 to 1121, with a mass resolution of $\Delta m/m = 2500$ to 3000.

The chemical composition of submicron aerosol particles was measured using a Compact Time-Of-Flight Aerosol Mass
Spectrometer (C-TOF-AMS) (Drewnick et al., 2005; Schulz et al., 2018). The C-TOF-AMS uses a constant pressure inlet (Molleker et al., 2020) followed by an aerodynamic lens to sample particles in a size range of between about 40 and 800 nm from ambient air and then focusses the particle beam onto a tungsten vaporiser, operated at 600 °C. The non-refractory



components of the particles are vaporised and the gas molecules are ionised by electron impact. The ions are then extracted
into the time-of-flight mass spectrometer and are detected by a multichannel plate detector. During CAFE-EU/BLUESKY, the
C-TOF-AMS was operated at a time resolution of 30 s. The detection limits lie in the range of between 20 and 110 ng m$^{-3}$,
depending on the species measured (Schulz et al., 2018).

Particle size distributions for particles larger than 250 nm in diameter were measured using an Optical Particle Spectrometer
(Grimm 1.129 "Sky-OPC"). The nominal size range of this instrument ranges from 250 nm to 32 µm, divided into 31 size
channels. Here, the upper size limit was determined by the aerosol inlet of HALO, whose upper size cut lies around 5 µm. The
time resolution was 6 s.

The relative humidity was measured by the Sophisticated Hygrometer for Atmospheric ResearCh (SHARC), which employs
a tuneable diode laser (TDL) system. Within clouds, the measurement of the relative humidity can be influenced by the
evaporation of cloud particles, thus, relative humidities exceeding 100 % are possible. Basic meteorological and flight
parameters, such as temperature, aircraft position and altitude, were measured by the BAsic HAlo Measurement And Sensor
system (BAHAMAS). More information on the instrumentation of the aircraft during the CAFE-EU/BLUESKY campaign is
given in the overview publication by Voigt et al. (2021).

## 2.2 Measurement flights

**Table 1: Details for all flights in which measurement in the APi mode were conducted.**

| Flight no. and segment | Mode | Date | Time [UTC] | Duration [min] | Latitude [° N] | Longitude [° E] | Altitude [km] |
|---|---|---|---|---|---|---|---|
| 04.1 | negative | 30 May 2020 | 08:08–09:37 | 89 | 50.3...52.2 | –13.3…9.8 | 11.1...11.9 |
| 04.2 | negative | 30 May 2020 | 13:52–15:03 | 71 | 50.5...52.2 | –9.7…4.6 | 10.9...11.0 |
| 05.1 | negative | 02 June 2020 | 09:04–09:55 | 51 | 50.3...52.2 | –13.3…–3.5 | 12.2...12.3 |
| 05.2 | negative | 02 June 2020 | 13:23–14:00 | 37 | 51.6...52.9 | –8.2…–0.5 | 13.4...13.5 |
| 05.3 | negative | 02 June 2020 | 14:43–15:11 | 28 | 48.6...49.9 | 7.8…10.8 | 4.7...9.7 |
| 06.1 | negative | 04 June 2020 | 08:25–08:41 | 16 | 46.6...47.5 | 6.5…8.8 | 11.3 |
| 06.2 | negative | 04 June 2020 | 10:59–11:06 | 7 | 43.0...43.4 | 9.6…10.1 | 5.3 |
| 07.1 | negative | 06 June 2020 | 08:16–08:32 | 16 | 47.1...47.7 | 7.2…9.3 | 10.2 |
| 07.2 | negative | 06 June 2020 | 09:38–09:53 | 15 | 43.7...44.9 | –0.4…1.2 | 10.2...10.3 |
| 08.1 | negative | 09 June 2020 | 08:21–08:54 | 33 | 45.4...47.4 | 4.0…8.4 | 9.2 |
| **all negative** | **negative** | | | **363** | **43.0...52.9** | **–13.3…10.8** | **4.7...13.5** |
| 08.2 | positive | 09 June 2020 | 14:31–15:57 | 86 | 47.1...48.5 | 8.0…12.4 | 0.6...12.7 |



The aforementioned instruments, amongst others, were installed on board the HALO research aircraft for the CAFE-
EU/BLUESKY campaign. During the campaign, one test flight and eight scientific flights were performed between 21 May
and 9 June 2020, i.e. during the COVID-19 lockdowns in most European countries. All flights commenced and terminated at
Oberpfaffenhofen airport (Bavaria, Germany). The area covered included Germany, the Netherlands, Switzerland and Italy
(flights no. 01 to 03), the North Atlantic Flight Corridor, including Ireland and the United Kingdom (04 and 05), as well as

France, Italy, and Spain bordering the Mediterranean Sea (06 to 08). Details of the campaign are given in the overview
publication by Voigt et al. (2021).

During the eight flights, the SCORPION-TOF-MS was operated in the Chemical Ionisation mode for the majority of the time.
During flights no. 04 to 08, measurements in the APi mode were performed for at least two periods for each flight (called
segments). In the negative APi mode, we took data for a total period of 363 min (6 h), covering an altitude range of 4.7 to

13.5 km, a latitude range of 43.0 to 52.9° N, and a longitude range of –13.3 to 10.8° E. In the positive mode, we performed
one measurement for 86 min, covering an altitude range of 0.6 to 12.7 km, a latitude range of 47.1 to 48.5° N and a longitude
range of 8.0 to 12.4° E. The details for all APi measurements are summarised in Table 1. For a better overview, the flight
tracks of all APi measurements are shown in Fig. 1.

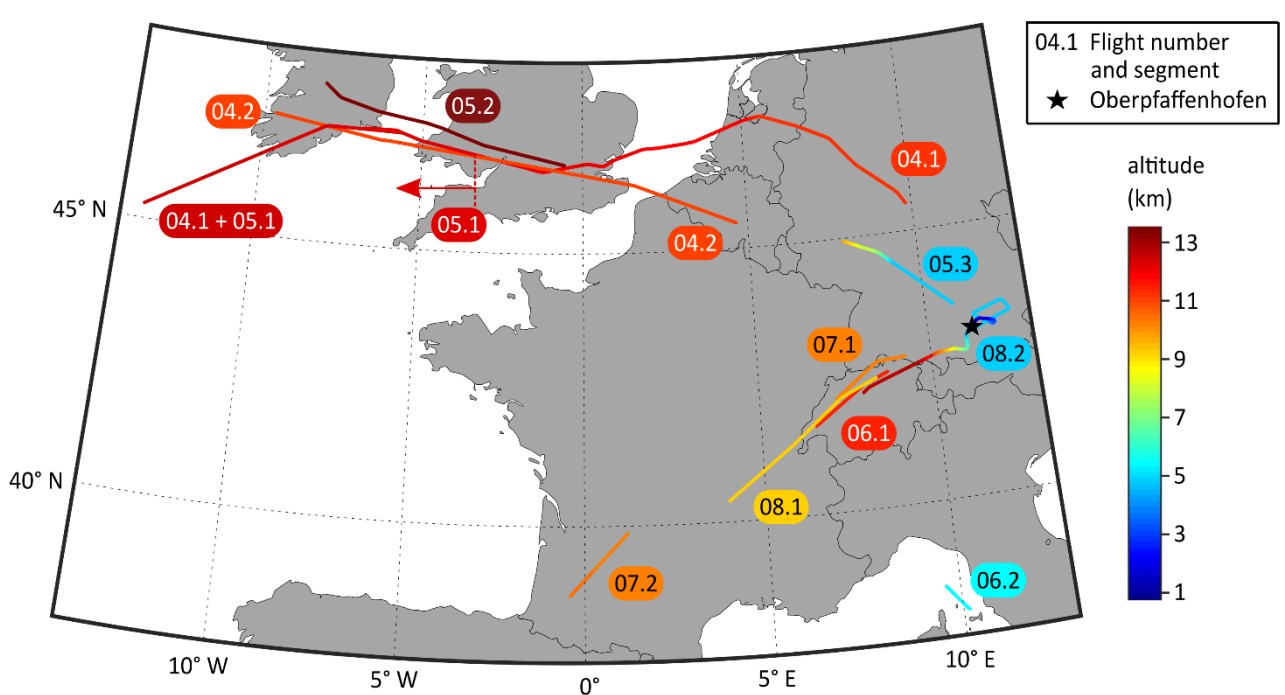

**Figure 1: Map showing the flight tracks during the taking of measurements in the APi mode. The colour code indicates the flight
altitude. The labels indicate the flight number and segment. The star represents Oberpfaffenhofen airport. National border data
obtained from Greene et al. (2019).**



## 2.3 Data analysis

The recorded mass spectrometric data were post-processed using the IGOR-based *Tofware* software provided by Aerodyne
Research Inc. Corrupted data caused by interference of the data acquisition unit with mobile and Wi-Fi radiation were removed.
This interference occurred mainly while HALO was on the apron and only affected < 0.2 % of the data obtained during each
flight (except flight 01: 1 % of the data). The uncorrupted data were then averaged to 30 s. The subsequent post-processing
included a detailed mass calibration, peak identification and peak integration. The integrated peak values were normalised by
the total ion count.

In the laboratory, background measurements of pure synthetic air were conducted. Reproducible peaks occurred at certain
mass-to-charge ratios $m/z$ in a range of up to 80, with 16, 26, and 19 being the most dominant ones in the negative mode. These
background peaks were likely caused by internal chemical processes in the mass spectrometer as they are dependent on the
sample gas. These peaks could also be found in the experimental spectra of the in-flight measurements. As their relative
abundance was roughly constant, they were subtracted from the recorded mass spectra of the measurement flights. The only
notable overlap with observed ambient ions in the negative mode occurred at nominal mass 62; the ambient ion signals were
much stronger than the laboratory background signals at nominal mass 62. The laboratory background signal was 5 % of the
total recorded signal at $m/z = 62$ in the highest altitude bin (13.3 km), whilst it constituted 15 % of the total recorded signal in
the lowest altitude bin (4.7–5.3 km). These values are within the general measurement uncertainty. Attempting to subtract a
background-related portion from the signal recorded during the flights may be detrimental to the results and, therefore, this
process was omitted.

## 2.4 Quantification of gaseous sulfuric acid

Based on Reaction (R1), the number concentration of gaseous sulfuric acid was calculated, using the ratio of $NO_3^-$ and $HSO_4^-$
count rates. This method was developed by Arnold and Fabian (1980) and is described in more detail by Arnold and Qiu
(1984). The number concentration of sulfuric acid can be derived from Eq. (1a):

$$H_2SO_4 + NO_3^- \rightarrow HSO_4^- + HNO_3 \tag{R1}$$

$$[H_2SO_4] = \frac{1}{k \cdot t_{rec}} \cdot \frac{CR(HSO_4^-)}{CR(NO_3^-)}, \tag{1a}$$

where $CR(HSO_4^-)$ and $CR(NO_3^-)$ represent the instrument's count rates of $HSO_4^-$ core ions (i.e. $HSO_4^-$, $(HNO_3)HSO_4^-$, and
$(H_2SO_4)HSO_4^-$) and $NO_3^-$ core ions (i.e. $NO_3^-$, $(HNO_3)NO_3^-$, and $(HNO_3)_2NO_3^-$), respectively, in counts per second (cps); $k$,
the reaction rate constant, is $2 \cdot 10^{-9}$ cm$^3$ s$^{-1}$ (Viggiano et al., 1997); and $t_{rec}$ is the ion-ion recombination lifetime in s. Equation
(1a) represents the special case, when $CR(HSO_4^-) << CR(NO_3^-)$. This is not always the case, thus, the more accurate
logarithmic term according to Eq. (1b) is used (Heinritzi et al., 2016):

$$[H_2SO_4] = \frac{1}{k \cdot t_{rec}} \cdot \ln\left(1 + \frac{CR(HSO_4^-)}{CR(NO_3^-)}\right). \tag{1b}$$

The ion-ion recombination lifetime can be calculated with Eq. (2):



$$t_{\text{rec}} = \frac{n_+}{q}, \tag{2}$$

where $n_+$ is the number concentration of positive ions and $q$ is the ion pair production rate. To calculate $n_+$, Eq. (3) can be used (Franchin et al., 2015):

$$\frac{\mathrm{d}n_\pm}{\mathrm{d}t} = q - \alpha n_+ n_- - k_{\text{CS}} n_\pm, \tag{3}$$

where $\alpha$ is the ion-ion recombination coefficient, and $k_{\text{CS}}$ is the condensation sink coefficient. Assuming $n_+ \approx n_-$ and that the term for the condensation sink is negligible due to little aerosol surface in the UTLS for ions to condense onto, Eq. (3) can be

simplified to Eq. (4):

$$\frac{\mathrm{d}n_\pm}{\mathrm{d}t} = q - \alpha n_+^2. \tag{4}$$

For steady-state conditions, Eq. (5) is attained (Thomson and Rutherford, 1896):

$$n_+ = \sqrt{\frac{q}{\alpha}}. \tag{5}$$

$\alpha$ is given as $1.6 \cdot 10^{-6}$ cm$^3$ s$^{-1}$ for conditions at sea level in today's literature (Franchin et al., 2015). This value is taken from

Israël (1957, 1971). However, $\alpha$ is temperature- and pressure-dependent and must be adjusted when used for altitudes above 10 km. In a separate work, we discussed different parameterisations and models of the ion-ion recombination rate and concluded that the parameterisation of Brasseur and Chatel (1983) is favourable (Zauner-Wieczorek et al., 2021); this is characterised by Eq. (6):

$$\alpha = 6 \cdot 10^{-8} \cdot \sqrt{\frac{300}{T}} + 6 \cdot 10^{-26} \cdot [\text{M}] \cdot \sqrt[4]{\frac{300}{T}}, \tag{6}$$

where $\alpha$ is in cm$^3$ s$^{-1}$, $T$ is the temperature in K, and [M] is the number concentration of air molecules in cm$^{-3}$ derived from the ideal gas law, described by Eq. (7), after Arijs et al. (1983):

$$[\text{M}] = 7.243 \cdot 10^{18} \cdot \frac{p[\text{hPa}]}{T[\text{K}]}. \tag{7}$$

Notably, this parameterisation yields an almost constant value of approximately 1.7 to $1.9 \cdot 10^{-6}$ cm$^3$ s$^{-1}$ for $\alpha$ from ground level up to 11 km when the values for the US Standard Atmosphere (National Oceanic and Atmospheric Administration et al.,

1976) are used; this is close to the above-mentioned value of $1.6 \cdot 10^{-6}$ cm$^3$ s$^{-1}$. Above 11 km, the value decreases rapidly, with $\alpha = 0.96 \cdot 10^{-6}$ cm$^3$ s$^{-1}$ at 15 km altitude.

The ion production rate $q$ is dependent on the altitude, the geomagnetic latitude and the solar activity (Bazilevskaya et al., 2008). While a typical value for $q$ at sea level is 2 ion pairs cm$^{-3}$ s$^{-1}$ (Hensen and van der Hage, 1994), the ion production rate increases with increasing altitude until it reaches a maximum value of 37 ion pairs cm$^{-3}$ s$^{-1}$ at 11 to 14 km for polar latitudes

(Bazilevskaya et al., 2008). However, as the ionisation by GCRs is stronger at the poles than at lower latitudes, the geomagnetic latitude must be taken into account. This can also be resembled by the geomagnetic cut-off rigidity, $R_C$, representing the shielding effect of the Earth's magnetic field (Smart and Shea, 2005). For the latitude range we covered during the measurement flights (43 to 53° N), a geomagnetic cut-off rigidity $R_C$ of 2 to 6 GV must be applied (Smart and Shea, 2009), therefore, the value for $q$ applied here must be 90 % of the maximum polar value of $q$ (Bazilevskaya et al., 2008). Furthermore,





solar activity influences the ion production in the atmosphere. Strong solar activities lead to deflection of GCRs and, consequently, a weaker ionisation; this is especially the case at an altitude of 11 km where the modulation of $q$ is highly dependent on the 11-year solar cycle, leading to a 20 to 25 % difference in $q$ between solar activity maxima and minima (Bazilevskaya et al., 2008). In May and June 2020, the solar activity was at a minimum (Royal Obervatory of Belgium, 2020), thus, maximum values for $q$ had to be considered. Based on the conditions during our sampling (elaborated above) and the

dependencies of $q$ summarised by Bazilevskaya et al. (2008), we estimated an ion pair production rate of 30 ion pairs cm$^{-3}$ s$^{-1}$ for our data.

## 3 Results and Discussion

### 3.1 Negative ions

#### 3.1.1 Mass spectrum

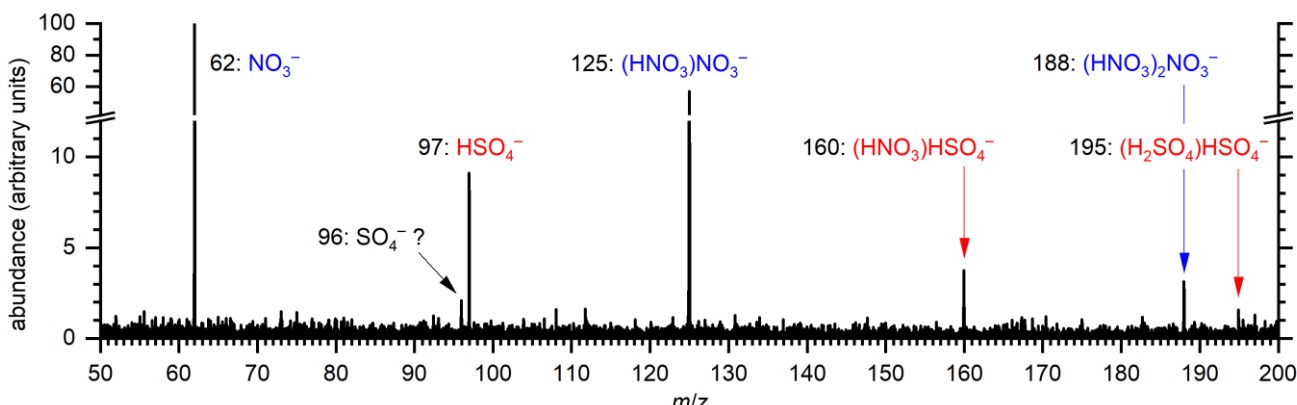


**Figure 2: Mass spectrum of ambient negative ions averaged for all measurements in the negative APi mode. Ions with a nitrate core ion are labelled in blue; those with a hydrogen sulfate core ion are labelled in red.**

Figure 2 shows the average mass spectrum for all measurements in the negative APi mode. The most dominant peak was at $m/z = 62$ and was assigned to nitrate, $NO_3^-$ (monomer), followed by the dimer cluster of nitric acid with nitrate, $(HNO_3)NO_3^-$

($m/z = 125$). In addition, the trimer cluster $(HNO_3)_2NO_3^-$ (188) could be observed. As expected in the UTLS, hydrogen sulfate, $HSO_4^-$ (97) was detected as well as its clusters with nitric acid (160, $(HNO_3)HSO_4^-$) and sulfuric acid (195, $(H_2SO_4)HSO_4^-$). Evidence for further clusters with $NO_3^-$ or $HSO_4^-$ core ions could not be retrieved from the mass spectrum, as either their concentrations are too low to exceed the background noise or they experienced fragmentation in the APi section of the instrument and were, thus, detected as one or more of the above mentioned ions. The peak at nominal mass 96 could not be

assigned to $SO_4^-$ for certain due to the measured exact $m/z$ value being 95.973, compared to the mass of 95.952 of $SO_4^-$. Water clusters such as $(H_2O)NO_3^-$ (80) could not be observed. The detected peak at nominal mass 80 was also present in the





background laboratory measurements and did not exceed the level of the background measurements during the in-flight measurements. The observed ions and their exact masses are listed in Table 2.

**Table 2: Observed signals in the negative mass spectra, their exact mass-to-charge ratio, *m/z*, and assigned ions.**

| *m/z* | Ion | Remarks |
|---|---|---|
| 61.988 | $NO_3^-$ | |
| 95.973 | $SO_4^-$ (?) | exact mass of $SO_4^-$: 95.952 Da |
| 96.960 | $HSO_4^-$ | |
| 124.984 | $(HNO_3)NO_3^-$ | |
| 159.956 | $(HNO_3)HSO_4^-$ | |
| 187.980 | $(HNO_3)_2NO_3^-$ | |
| 194.927 | $(H_2SO_4)HSO_4^-$ | |

The (almost) exclusive detection of the $NO_3^-$ or $HSO_4^-$ core ions in the ambient ion mass spectrum is typical for the UTLS region and is in accordance with previous measurements (Heitmann and Arnold, 1983). The most notable discrepancy from Heitmann and Arnold's results is the relative abundance of the signals. They found $(HNO_3)_2NO_3^-$ to be the most abundant species, followed by $(HNO_3)NO_3^-$ and $(HNO_3)_2HSO_4^-$, with only a slight amount of $NO_3^-$. In our findings, the smaller ions were more abundant than the bigger clusters. Unfortunately, the resolution was comparably low in the earlier studies, so that

the peak identified as the trimer (labelled with the *m/z* value 188±2) is wide enough to include, easily, other species with *m/z* ratios of approximately 170 to 205. Possible explanations for this discrepancy are different mass-dependent transmission efficiencies between Heitmann and Arnold's instrument and ours or the potential fragmentation of larger clusters in our instrument.

**3.1.2 Altitude dependence of the ions and quantification of sulfuric acid**

In Fig. 3 (a) and (b), the normalised count rates of $NO_3^-$ and $(HNO_3)NO_3^-$, respectively, are shown in dependence of the altitude in a box plot. The vertical distributions of the count rates of $NO_3^-$ and $(HNO_3)NO_3^-$ show an increasing trend with increasing altitude. The ion count rates of both ions are elevated above 11.5 km, i.e. elevated in the lower stratosphere compared to the free troposphere. While the values might not differ significantly from one altitude bin to another, the difference between, for instance, the 13.4 km and the 11.0–11.3 km bins is substantial. The average normalised count rates (squares in Fig. 3) for the

13.4 km bin are $5.1 \cdot 10^{-3}$ for $NO_3^-$ and $3.2 \cdot 10^{-3}$ for $(HNO_3)NO_3^-$. For the 11.0–11.3 km bin, the average count rates are about a factor of 3 lower ($1.6 \cdot 10^{-3}$ for $NO_3^-$ and $1.2 \cdot 10^{-3}$ for $(HNO_3)NO_3^-$). This finding is in accordance with the direct measurements of the corresponding neutral molecule, $HNO_3$, which has been found to be elevated in the stratosphere (Schneider et al., 1998; Schumann et al., 2000; Jurkat et al., 2016). For $HSO_4^-$, a clear altitude dependence cannot be retrieved from our data, as Fig. 3 (c) shows. The differences between two altitude bins can be as high as a factor of 2 (e.g. $2.2 \cdot 10^{-4}$ at

11.0–11.3 km, compared to $4.5 \cdot 10^{-4}$ at 10.2 km). The value of $0.97 \cdot 10^{-4}$ at 4.7–5.3 km is comparably small, although the number of data points below 8.3 km is low and, thus, the informative value should not be overestimated.

For every 30 second averaged data tuple of the respective sums of the count rates of the $NO_3^-$ core ions and the $HSO_4^-$ core ions, the number concentration of sulfuric acid was calculated according to Eq. (1b). From Eq. (5) and (2), we derived the number concentration of positive ions $n_+$ and the ion-ion recombination lifetime $t_{rec}$ for every data tuple. The average $t_{rec} =$ 290 130 s (107 to 161 s) and the average $n_+ = 3900$ cm$^{-3}$ (3220 to 4830 cm$^{-3}$). We checked the term for the condensation sink for each flight (see Eq. (3)), derived from the Sky-OPC data, and found that it was two orders of magnitude below the recombination term for each flight and that it was, thus, negligible, except for flight segment 06.2; this is addressed in more detail in the next subsection.

The results for the gaseous sulfuric acid number concentration, $[H_2SO_4]$, are shown as a box plot graph in dependence of the 295 altitude in Fig. 3 (d). The smallest average value of $1.8 \cdot 10^5$ cm$^{-3}$ $H_2SO_4$ was observed at the maximum altitude of 13.4 km. The averages increased with decreasing altitude up to $9.1 \cdot 10^5$ cm$^{-3}$ $H_2SO_4$ in the 8.7–9.2 km bin; below this altitude, the average concentrations became slightly lower again.

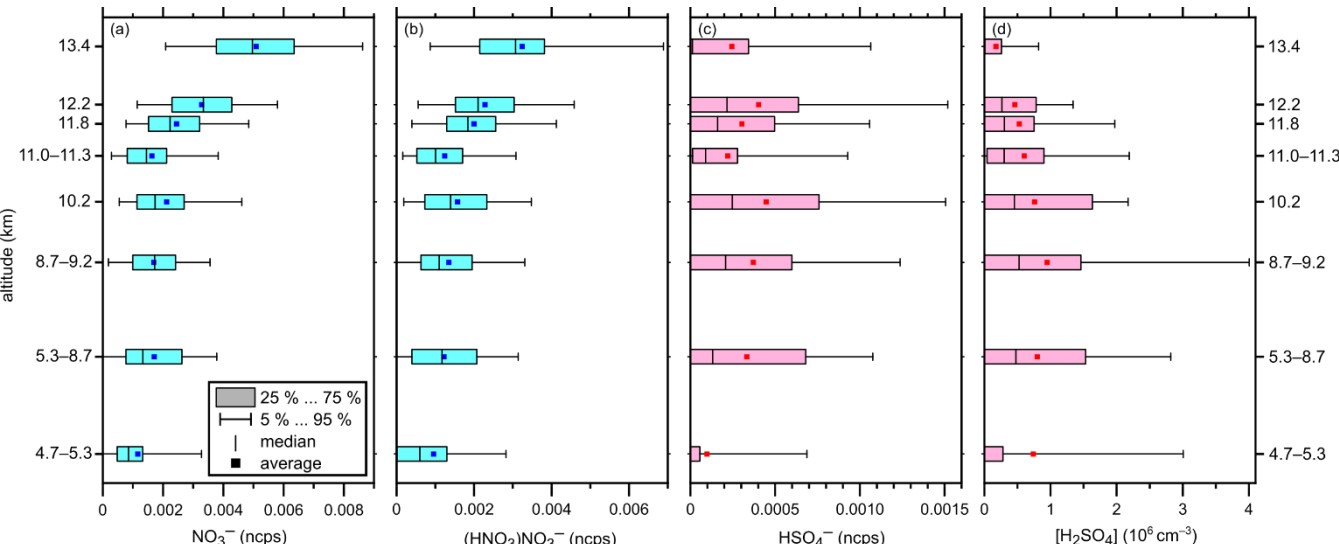

**Figure 3: Altitude-resolved box plots of the normalised count rates of (a) NO₃⁻, (b) (HNO₃)NO₃⁻, and (c) HSO₄⁻, and (d) the number**
**concentration of gaseous H₂SO₄. The units for (a) to (c) are normalised counts per second (ncps), i.e. the respective ion counts divided by the total ion counts. Please note the different x-axis ranges. The unit for (d) is cm⁻³. The boxes include all measured points between the 25 % and the 75 % percentiles, i.e. the boxes contain the medium half of the data points. The outer bars indicate the range of all data points except for the lowermost 5 % and uppermost 5 %. The vertical bars inside the boxes indicate the respective medians and the squares indicate the respective averages.**

If only the altitudes above 8.7 km, where the data coverage was better, are taken into account, the trend in average concentrations is in accordance to previous findings. Measurements conducted by Heitmann and Arnold (1983) in the altitude range of between 8 and 20 km above West Germany showed that the total number concentration of acidic sulfur gases, of which sulfuric acid is expected to be the most dominant, has a decreasing trend with increasing altitude, with the highest value



of $10^7$ cm$^{-3}$ being at the minimum altitude of 8 km, compared to $10^6$ cm$^{-3}$ at 12 km. It should be noted that the SO$_2$ emissions

were significantly higher in Europe in the 1980s compared to today. SO$_2$ is the most important precursor to H$_2$SO$_4$ (Stockwell and Calvert, 1983; Seinfeld and Pandis, 2006). As a rough estimate, one can assume an SO$_2$ emission reduction of 90 to 95 % from the mid-1980s to 2020 (Smith et al., 2011; European Environmental Agency, 2021). As SO$_2$ is the main source for sulfuric acid in the upper troposphere (Seinfeld and Pandis, 2006), the relatively high concentrations of sulfuric acid in the study by Heitmann and Arnold (1983) can be explained by this assumption. Möhler and Arnold (1992) reported H$_2$SO$_4$ mixing

ratios for the altitude range of 6 to 12 km in Northern Scandinavia during February 1987; the mixing ratio of sulfuric acid was slightly elevated in the tropopause region (2 to 6 · $10^{-14}$ around 9.5 to 10 km), otherwise it remained at a rather constant level (1 · $10^{-14}$). When converted to number concentrations, the data reproduce the pattern observed by Heitmann and Arnold (1983), i.e. decreasing H$_2$SO$_4$ number concentrations with increasing altitude. The resulting number concentrations are between 6 · $10^5$ cm$^{-3}$ at 12 km and 1 · $10^4$ cm$^{-3}$ at 6 km, and are, thus, much less than those reported by Heitmann and Arnold (1983);

this is because there are no local pollution sources in the Arctic, even though the sampled air masses have travelled there from lower latitudes, which are more prone to pollution (Möhler and Arnold, 1992). In both studies, gaseous sulfuric acid concentrations were inferred from the ambient ion measurements of NO$_3^-$ and HSO$_4^-$ using the same approach as we have done. In addition, the most recent model for tropospheric ion composition by Beig and Brasseur (2000), based on the observations from the 1980s and 90s, predicts a strong increase in HSO$_4^-$ core ions and H$_2$SO$_4$ below 15 km, with a maximum

at around 8 km. Furthermore, Mauldin et al. (2003) reported uniformly low sulfuric acid concentrations in the free troposphere of, typically, 1 to 8 · $10^5$ cm$^{-3}$ between 5 and 7.5 km altitude and usually 0.5 to 2 · $10^6$ cm$^{-3}$ between 3 and 5 km. Note that the only two reference studies for detecting sulfuric acid in the UTLS by use of ambient ion measurements were conducted under dissimilar conditions compared to our study: Möhler and Arnold (1992) studied air masses in the Arctic winter with very low concentrations of H$_2$SO$_4$, whilst the results of Heitmann and Arnold (1983) were influenced by massively larger SO$_2$ emissions

compared to today. Unfortunately, there are no other direct measurements of neutral sulfuric acid molecules in the UTLS reported in the literature. Thus, direct measurements of gaseous sulfuric acid in the UTLS by state-of-the-art chemical ionisation mass spectrometric techniques are needed for further comparison and validation of the ambient ion-inferred results.

### 3.1.3 In-cloud measurement

A unique observation within the CAFE-EU/BLUESKY data set was captured during the APi measurement 06.2, on 04 June

2020 at 11:02 UTC, when the aircraft flew through a cloud at a flight altitude of 5.3 km. During this period, an incident of 30 s in duration consisting of high ambient ion count rates, high aerosol particle concentrations, and high humidity occurred. In Fig. 4, the relevant parameters for this incident are depicted. While the total ion count rate increased from approximately 51 to 61 ions s$^{-1}$, the total nitrate count rate (i.e. the sum of NO$_3^-$, (HNO$_3$)NO$_3^-$, and (HNO$_3$)$_2$NO$_3^-$) increased from 0.2–0.4 to 5.3 ions s$^{-1}$, being responsible for half of the total ion count increase (see Fig. 4 (a)). Between 11:01:57 and 11:02:20 UTC,

the relative humidity over water showed three peaks of 132–136 % compared to 114 % before and after this event (see Fig. 4 (b)). The temperature, however, was constant at 275 K, only decreasing by up to 1 K during the humidity peak events and was,



thus, always above the freezing point. The aerosol mass concentration (Fig. 4 (c)) peaked at 11:02:30 (at a time resolution of 30 s) with a total of 3.2 µg cm$^{-3}$ (at standard temperature and pressure, STP). As in the case of the gas-phase ions, nitrate played an important role in this event, representing approximately 40 % of the particle mass. The lowest panel, Fig. 4 (d),

reveals the particle size distribution and number concentration. Shortly after 11:02:00, two bands of increased particle number concentrations can be observed that correlate well with the peaks of the humidity data. The concentrations of all particle sizes up to 5 µm diameter increased by a factor of up to 100 compared to before and after the event. Most of the aerosol particles were smaller than 1 µm during the event.

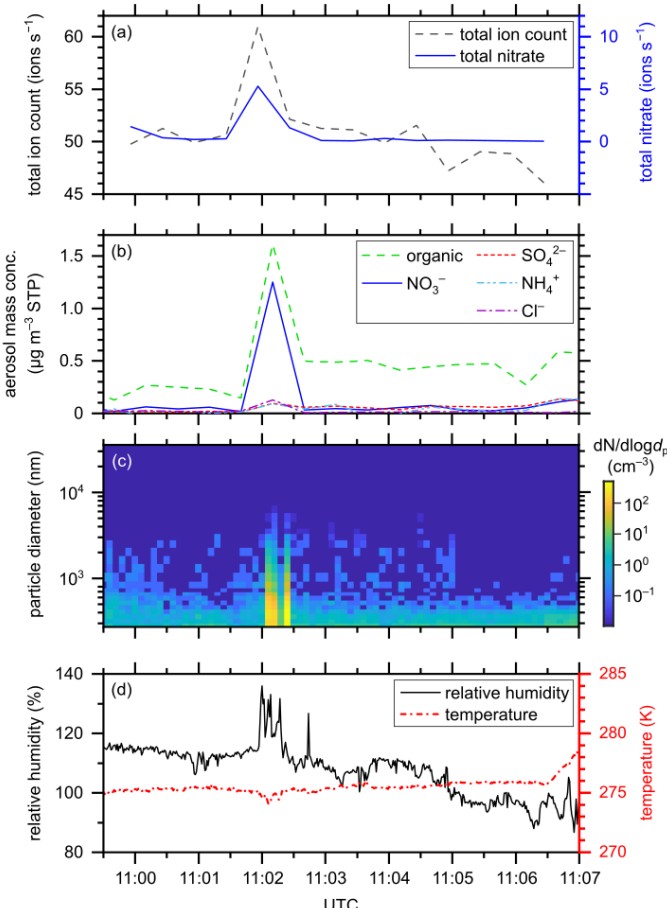

**Figure 4: Gas-phase and particle-phase parameters during the APi measurement on flight segment 06.2 on 04 June 2020 above the Mediterranean Sea at a flight altitude of 5.3 km. (a) Total ion count rate (dashed black curve) and total nitrate count rate (solid blue curve) in ions s$^{-1}$ provided by the APi-TOF-MS, (b) aerosol mass concentration in µg per standard m$^3$, indicated by the chemical composition, provided by the C-TOF-AMS, (c) aerosol particle size distribution in nm and number concentration in cm$^{-3}$ provided by the Sky-OPC, and (d) relative humidity over liquid water in % (solid black curve) provided by the SHARC, and temperature in**
**K (dashed-and-dotted red curve) provided by the BAHAMAS.**

Since the aircraft passed through a liquid cloud during this event, it is likely that the shattering of liquid cloud droplets on the surface of the aircraft or the sampling system caused or at least contributed to the observed increase in aerosol mass





concentration. We speculate that this observation can be explained by the balloelectric effect (Christiansen, 1913) which describes the generation of electric charge by the shattering of water droplets: in the process of water shattering on the aircraft or inlet surface, the concentration of ions increased; the subsequent reaction time in the inlet system was, however, too short for nitrate to react with other species in order to transfer its charge. It is known that nitric acid and organics are favourably taken up by liquid cloud droplets (e.g. Schneider et al., 2017), which can explain the dominance of nitrate and organics in the measured data. However, the effect of shattering cloud droplets on the ambient ion concentration is not straightforward, such that the confident interpretation of these results requires further in-cloud measurements in the APi mode in the future.

## 3.2 Positive ions

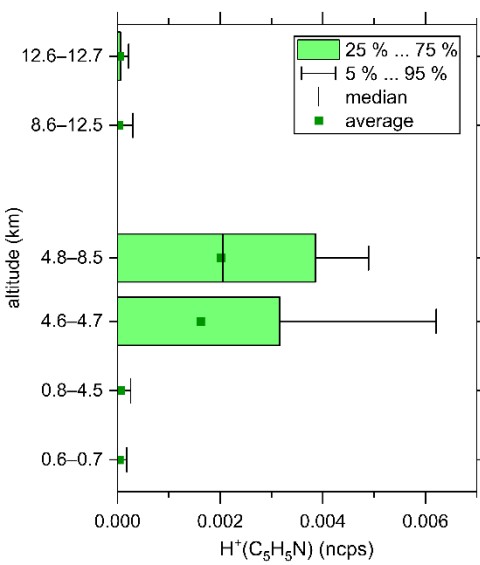

**Figure 5: Altitude-resolved box plot of the normalised count rates of $H^+(C_5H_5N)$. The boxes include all measured points between the 25 % and the 75 % percentiles, i.e. the boxes contain the medium half of the data points. The outer bars indicate the range of all data points except for the lowermost 5 % and uppermost 5 %. The vertical bars inside the boxes indicate the respective medians and the squares indicate the respective averages.**

We recorded the mass spectra of positive ambient ions for a total of 86 min during the end of the last measurement flight at an altitude range of 0.6 to 12.7 km. This data coverage does not allow for a detailed analysis of positive ambient ions as performed for the negative ions. However, we can provide the proof of principle for positive ambient ion analysis with our SCORPION-HR-TOF-MS device. We unambiguously detected protonated pyridine (80, $H^+(C_5H_5N)$), which was especially abundant between 4.6 and 8.5 km (see Fig. 5). Below 4.6 km and above 8.5 km altitude, almost no protonated pyridine was detected. This species was found to be the most abundant positive ion between 3 and 6 km in a previous study (Schulte and Arnold, 1990). There were also signals that strongly hinted to protonated acetonitrile (42, $H^+(C_2H_3N)$) and protonated acetone (59, $H^+(C_3H_6N)$), however, further studies with longer measurement durations must be performed to distinguish, reliably, those signals from the background and to detect and identify further positive ambient ions.



## 4 Conclusion


We have presented the first measurement of ambient ions in the free troposphere and lower stratosphere with an HR-APi-TOF-MS above Europe at an altitude range of 4.7 to 13.4 km. We identified the major ambient negative ions $NO_3^-$ and $HSO_4^-$ and their respective clusters with $HNO_3$ and $H_2SO_4$, confirming previously reported research. The nitrate core ions are dominant over the hydrogen sulfate ion family. For $(HNO_3)_{0,1}NO_3^-$ and $HSO_4^-$, the detailed altitude-resolved abundances are presented;

the nitrate core ions are elevated in the stratosphere, whereas hydrogen sulfate shows no significant altitude trend. From the ratio of hydrogen sulfate-to-nitrate core ions, we inferred the number concentration of gaseous sulfuric acid. The resulting average values were determined to be 1.8 to $9.1 \cdot 10^5$ $cm^{-3}$ for the altitude range of 4.7 to 13.4 km, with an overall observed decreasing trend towards higher altitudes. During the transit through a water cloud, we observed an event of concurrent elevated aerosol particle concentrations and ambient ion counts that may be explained by the shattering of cloud water droplets

on the surface of the airplane or inlet system. Nitrate played the dominant role in the chemical composition of both the measured ions and the particles. Furthermore, we provided a proof of principle for the measurement of positive ions and were able to unambiguously identify protonated pyridine ($H^+(C_5H_5N)$) which is especially abundant between 4.6 and 8.5 km altitude. For future research, more data need to be acquired in order to complete the altitude profiles of ambient ions, especially cations. In addition, up-to-date chemical ionisation mass spectrometric measurements of gaseous sulfuric acid are needed for the UTLS

in order to compare and validate the data presented herein.

### Author contribution

MZW, MH, and JC designed the study. MZW, MH, MG, TK, KK, and JS prepared and conducted the experiments. MZW analysed the data. MZW, MH, AK, and JC discussed the results. MZW wrote the manuscript with contributions from MH and JS. All co-authors provided input for revision before submission.

### Competing interests


The authors declare that they have no conflict of interest.

### Acknowledgements

We thank the Max Planck Institute for Chemistry and its staff for organising and conducting the CAFE-EU/BLUESKY campaign and the German Aerospace Center (DLR) and its staff for providing the infrastructure, especially during the Corona

lockdown.



**Financial support**

This research has been funded by the Deutsche Forschungsgemeinschaft (DFG, German Research Foundation) – TRR 301 "TP-Change" – project ID 428312742, research project A03 and has been supported by the Messer Foundation (Bad Soden am Taunus, Germany). Marcel Zauner-Wieczorek is funded by the Heinrich Böll Foundation.

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
