# Peer review of "Mass spectrometric measurements of ambient ions and estimation of gaseous sulfuric acid in the free troposphere and lowermost stratosphere during the CAFE-EU/BLUESKY campaign"

_Atmospheric Chemistry and Physics, 2022_

## Author Response (AR1)

**Referee 1**

(Comments by the referee are in bold font, answers by the authors are in regular font)

In the manuscricpt "Mass spectrometric measurements of ambient ions and estimation of gaseous sulfuric acid in the free troposphere and lowermost stratosphere during the CAFE-EU/BLUESKY campaign", M. Zauner-Wieczorek and co-authors present highaltitude measurements of ambient ion compositions over Europe using a state-of-the-art ion mass spectrometer (HR-APi-TOF-MS).

Airborne deployments of such equipment for detailed ion measurements are rare, even more so for high altitudes (ULTS). Given that rarity of the methodology and resulting data and their analysis, I believe the paper overall is novel enough to justify publication in ACP. But due to the limited scope of the results and their discussion, I would recommend publication rather as a "measurement report" than a research article.

The manuscript is well written overall. Worth pointing out in particular is the good summary of relevant past studies in the Introduction section, which puts the study into proper perspective.

I have a two major comments relating to instrument performance, data analysis and the estimation of sulfuric acid concentrations, which I believe should be addressed prior to acceptance.

**A couple of minor and technical comments are mostly suggestions that I hope would improve the paper further.**

We would like to thank the Referee for their valuable feedback. Thanks to their suggestions, we believe that the manuscript could be significantly improved.

As the Referee notes, ion measurements are rare for the UTLS, especially in the last few decades. Moreover, measurements of gaseous sulfuric acid in the UTLS have been very few as discussed in the introductory part of our manuscript. Thus, the data set of ion measurements and the inferred gaseous sulfuric acid concentrations presented in our manuscript are an important contribution in a scarce field of research. Therefore, we are convinced that the publication as a research article is justified.

**Major comment 1:**

Two issues came to mind that could affect the validity of this method for estimating gaseous sulfuric acid concentrations, but aren't mentioned (Equations 1).

One being the ion transmission as a function of mass (or m/z), which appears here is implicitly assumed to be the same for NO3- and HSO4-. Do the authors have any estimate on that transmission function, or how it could affect the outcome of Eqs. 1?

Heinritzi et al. (2016) studied the mass-dependent transmission of the very same HR-TOF-MS that we also used in this study. They used an Eisele-Tanner type corona discharge chemical ionisation source (Kürten et al., 2011) that was operated at the ambient pressure of around 1000 hPa. In that study, Heinritzi et al. (2016) found that at around m/z = 100, the transmission varies between 0.8 and 1.5 %. The most prominent differences to the former setup and the setup used in this study are the different inlet lines, the two critical orifices between the inlet line, the mid-pressure stage, and the ion source in the SCORPION setup, the lower pressure in the ion source of the SCORPION, and the different internal voltage settings of the TOF-MS. Beck et al. (2022a), who used a similar\* *Tofwerk* HR-TOF-MS, found very similar values for the transmission. [\* See the question in the minor comments about the mass spectrometer used by Junninen et al. (2010), which was also used by Beck et al. (2022a).]

Thus, a conservative estimate for the transmission difference between  $NO_3^-$  and  $HSO_4^-$  leads us to a scaling factor of maximum 2. This factor is within our overall experimental uncertainty.

For the future, we plan to perform a more detailed characterisation of the mass-dependent transmission via the HR-DMA method described by Heinritzi et al. (2016). Due to time restrictions and campaign planning, we could not conduct this so far. This will be subject of a technical paper on the SCORPION by Heinritzi et al. (in preparation).

Also, I believe that some possibly important sinks of NO3- and HSO4- ions are being neglected by only considering R1. I am thinking specifically of adduct formation (with neutral molecules) that these ions are known to contribute to - potentially "consuming" a considerable portion of NO3- and/or HSO4-. Major candidates, also in the free troposphere, could be HNO3 and H2SO4, as well as organic acids. See several papers cited (e.g., Frege et al., 2017, 10.5194/acp-17-2613-2017; Ehn et al., 2010, 10.5194/acp-10-8513-2010) and also the data presented in the manuscript itself (e.g., Fig. 2). Could the authors elaborate on the role of such clusters for the accuracy of Eqs. 1?

If I missed here something explained in the 1980s papers, it could be helpful to summarize the key points of those studies regarding Eqs. 1, as many (myself included) do not have free access to many of those older studies.

We are sorry for the misunderstanding. Eq. (1a) and (1b) already include the clusters of  $NO_3^-$  and  $HSO_4^-$  with  $HNO_3$  and  $H_2SO_4$ . The ions/ion clusters  $NO_3^-$ ,  $(HNO_3)NO_3^-$ , and  $(HNO_3)_2NO_3^-$ ) are called nitrate core ions or  $NO_3^-$  core ions. The ions/ion clusters  $HSO_4^-$ ,  $(HNO_3)HSO_4^-$ , and  $(H_2SO_4)HSO_4^-$  are called hydrogen sulfate core ions or  $HSO_4^-$  core ions.

We calculated  $[H_2SO_4]$  for both the ratio of  $HSO_4^-/NO_3^-$  only and for the ratio of the respective sum of the core ions and found that the difference is not significant. This can well be explained by the fact that, in our data set, the count rates of  $NO_3^-$  and  $HSO_4^-$  are higher than the count rates of the clusters and, thus, the unclustered ions dominate the calculation using all core ions.

We changed the introduction of Sect. 2.4 and the explanation of Eq. (1a) slightly, in order to avoid misunderstandings and to include the key points of the derivation of Eq. (1a) by Arnold and Qiu (1984):

p. 8, l. 201ff.: "To quantify the number concentration of gaseous sulfuric acid, we used the steadystate method developed by Arnold and Fabian (1980), which is described in more detail by Arnold and Qiu (1984). This is based on the assumption that hydrogen sulfate ions are virtually only produced by charge transfer from nitrate to sulfuric acid, as given in Reaction (R1):

**[Reaction (R1)]**

Nitrate and hydrogen sulfate ions cluster with HNO3 and H2SO4 ligands, yielding (HNO3)mNO3-, called nitrate core ions, and (H2SO4)k(HNO3)mHSO4-, called hydrogen sulfate core ions, respectively (see also Sect. 1). Assuming negligible other source reactions for HSO4- core ions, a negligible aerosol sink, and subsequent ion-ion recombination after the reaction, Arnold and Qiu (1984) presented Eq. (1a) to calculate the number concentration of sulfuric acid from the ratio of the product ions (i.e. the HSO4- core ions) and the precursor ions (i.e. the NO3- core ions) for steady-state conditions:

[Eq. (1a)]"

**And sub-comment 4: Have the authors attempted the method suggested by Beck et al. (2021a; brought up in this study's introduction)? Theirs was somewhat simplified too, but differently.**

In the method presented by Beck et al. (2022a), the number concentration of gaseous sulfuric acid is calculated using the concentrations of the sulfuric acid monomer, dimer, and trimer. Because we barely observe the dimer (see Fig. 2 of the manuscript) and we are unable to detect the trimer in our data, it is, unfortunately, not possible to derive the sulfuric acid concentration with the method proposed by Beck et al. for our data set.

**Major comment 2**

Overall, are the authors able to comment on the ion transmission of their instrument (either as deployed, or as expected from pre- or succeeding experiments)? This comment relates to my comment above, but I am also thinking about the instrument's sensitivity overall. The signal intensity appears to be quite low here, whereas other studies, using a similar base mass spectrometers, appear to have obtained richer, less noisy spectra (e.g., Ehn et al., 2010, 10.5194/acp-10-8513-2010; Frege et al., 2017, 10.5194/acp-17-2613-2017).

Is that so, or just appearing that way to this reviewer? For example, do ground-based deployments mainly benefit from being able to average over longer measurement times? But Table 1 suggests that substantial averaging times were in fact available here. Also the airborne APi-TOF-based data in Beck et al. (2021, under review, 10.5194/acp-2021-994) appears to have "richer" spectra, at around 10 min averaging it seems. Though

**they do seem less rich in the instances where they measured in the (lower) free troposphere.**

**If this is all just how things are in the UTLS: Have the authors gathered any experience of how the SCORPION/APi-TOF performs near the surface (or even on the ground)?**

The signal intensity of our data set is lower and the mass spectrum less rich compared to other APi-TOF-MS studies at ground or in the lower free troposphere for several reasons:

- The abundance of ions is generally higher in the UTLS compared to the boundary layer because of the stronger exposition to galactic cosmic rays (GCRs). However, as described in the introductory section of the manuscript, in the UTLS there are virtually only NO3- and HSO4- anions and their clusters with HNO3 and H2SO4, which explains the less rich mass spectrum.
- The limited measurement times in the APi mode at different altitudes constitute a disadvantage compared to ground-based, locally fixed measurements where averaging times can be chosen more generously. Nevertheless, the boxplots in Fig. 3 and 5 are based on the 30 s averaged data points.
- Due to the specific setup in the aircraft, there are unavoidable losses along the way towards the APi-TOF-MS. The inlet line is 1.7 m long and there are two critical orifices that are necessary to ensure a constant pressure of 200 hPa inside the "ion source" (in the APi mode, the ion source region merely functions as a pressure controlled pre-chamber before entering the first vacuum chamber of the APi-TOF-MS). These lead to inevitable losses and, thus, a decreased sensitivity.
- Furthermore, the internal voltage settings of the APi-TOF-MS, including the multichannel plate (MCP) detector, were not optimal during the CAFE-EU/BLUESKY campaign, which was our first aircraft-based campaign with this setup. Measurements of ambient ions near the surface showed that with optimised internal voltage settings, the signal-to-noise ratio could be improved and the de-clustering of cluster ions could be reduced. We were able to clearly distinguish the dimers (HNO3)NO3- and (H2SO4)HSO4- from the background noise even for shorter averaging times (e.g. half an hour). We are looking forward to future measurement campaigns with optimised voltage settings.

In that respect, Fig. 2 could benefit from stating also in the caption, how long of a total average is shown. Line 253 suggests it is even a campaign-average, which would mean a total averaging time of 6 hours (if I understand correctly, and using Table 1)!? And not, for instance, an example of a 30-s spectrum, in which case the signal-to-noise would actually be quite good, on first glance ... but then, on the other hand, I would be rather surprised at the bleakness of the spectra, i.e. absence of additional peaks appearing with longer averaging (unless that was a feather of the UTLS)...

There is also a "minor comment" on Section 2.1 below that relates to this issue.

Figure 2 indeed shows a campaign-wide averaged mass spectrum. For more clarity, we added the averaging duration to the caption:

p. 11, l. 273f.: "Figure 2: Mass spectrum of ambient negative ions averaged for all measurements in the negative APi mode (i.e. averaged over 6 h). ..."

**Minor comments:**

**Line 27:**

Does the presence of ions in ambient air really qualify the atmosphere as a "plasma" (even a "diluted" one)? For example, I wouldn't consider a dust storm a "diluted solid"...

We found the perspective of a high energy physicist thought-provoking to view Earth's atmosphere as a "diluted plasma". However, this concept does not contribute to the actual topics discussed in the manuscript so that we deleted this term to avoid confusion:

p. 2, l. 27f.: "Earth's atmosphere not only contains neutral gases, but also ions in the gas phase that play a crucial role in several atmospheric processes..."

**Line 53: Could add recent paper by Beck et al. (2021; still in ACPD it appears; doi 10.5194/acp-2021-994).**

We added the paper by Beck et al. (2022b).

Lines 62-74: The paragraph may give the impression that vertically resolved nitric acid measurements are rarer than they are. Airborne nitric acid mixing ratio measurements go back at least to the PEM campaigns in the 1990s (Hoell et al., 1997, 1999; doi: 10.1029/97JD02581, 10.1029/1998JD100074). The most sensitive recent methods may be CIMS, e.g. using I- or CF3O-, deployed on aircraft on multiple occasions (e.g., Lee et al., 2018, doi 10.1029/2017JD028082; Dörich et al., 2021, doi 10.5194/amt-14-5319-2021). I suggest slightly extending that part of the introduction accordingly.

We extended the part about HNO3 measurements accordingly:

p. 3, l. 74ff.: "Early airborne measurements of nitric acid were performed in the 1990s (Hoell et al., 1997; Hoell et al., 1999). Iodide-adduct Chemical Ionisation Mass Spectrometry (CIMS) is one of the most sensitive methods used nowadays for airborne HNO3 detection (Lee et al., 2018; Dörich et al., 2021). Measurements in the upper troposphere and lower stratosphere (UTLS) showed that..."

**Section 2.1:**

It is not unambiguously clear that the base mass spectrometer is (presumably!) identical to that first described in Junninen et al. (2010), as introduced in the Introduction.

The hardware of the MS is identical to the one used by Junninen et al. (2010), however, the pumping architecture is different so that we are hesitant to call both base mass spectrometers identical.

In addition, it could be instructive to provide more detail regarding how operation with the SCORPION inlet, even when rendered inert with all voltages disabled, differed from the "default" operation of the TOF-MS to measure atmospheric ions. And in that respect, have the authors tried to quantify ion transmission losses due to the presence of the SCORPION inlet?

**All these could be of interest for readers engaged (or thinking about) ambient ion measurements using (the Tofwerk) TOF-MS.**

To switch from Chemical Ionisation (CI) mode ("default" mode) to Atmospheric Pressure inlet (APi) mode, only the internal voltages of the ion source, including the corona voltage, needed to be switched off. The gas flows inside the ionisation region were kept on because the contamination of  $HNO_3$  was excluded by implementing a counter flow regime with a separate exhaust flow close to the corona needle.

In CI mode, we have performed experiments to quantify transmission losses due to the inlet line, however, we have not yet fully analysed those data sets. Using the Gormley-Kennedy equation (Gormley and Kennedy, 1949), we estimate a loss of around 50 % for gaseous sulfuric acid. Losses inside the pressure stage are covered by the experimentally determined calibration factor when using the CI mode.

With regards to the ambient ion measurements presented in this manuscript, we assume that the difference in the wall losses to the inlet line and the critical orifices between  $NO_3^-$  and  $HSO_4^-$  are negligible. Because we only report relative values for the atmospheric ions and use the ratio of nitrate and hydrogen sulfate core ions, this does not affect the inferred concentration of sulfuric acid.

**Line 138: Any particular reason, why the mass resolution was "only" up to 3000? I recall somewhat higher numbers at least for Tofwerk's "H-TOF" (which I imagine was used).**

The mass resolution depends on the mass range and is typically in the range of 3000 for m/z values around 100 and increases with increasing m/z, experiencing a plateau of around 4000 for m/z > 300. The mass resolution also depends on the internal voltage settings of the TOF-MS. The official value given by the producer may be higher, however, the real values in the lab or in the field often differ from the ones given by the producer.

**Also, how good of a mass accuracy (ppm) was achieved?**

The mass accuracy was around 5 ppm for NO3-.

**Section 2.3, 1st paragraph: Was some sort of noise reduction necessary as well, or was the noise negligible compared to total ion counts (given there was no ionization source)?**

First of all, we average the 1 s raw data files to 30 s to diminish the noise. Afterwards, we performed a baseline subtraction as a standard procedure to process the averaged data, i.e. subtracting the average electronic noise from the mass spectra.

We added the information on the baseline subtraction to the manuscript:

p. 8, l. 186f.: "The subsequent post-processing included a detailed mass calibration, baseline subtraction, peak identification, and peak integration."

**Line 187: Do the authors mean ''INdependent''? Then I would understand that part better, I think. Else, please clarify.**

These background peaks appeared at different m/z ratios, depending on what sample gases we used. For synthetic air, peaks were observed that can be assigned to O-, O2-, N-, and N2-, amongst others. When using pure nitrogen, the O-related signals were not detectable. With argon, none of the aforementioned ions were detectable, but we observed Ar+ in the positive mode, instead.

We specified the sentence in the manuscript for more clarity:

p. 8, l. 190ff.: "These background peaks were likely caused by internal chemical processes in the mass spectrometer as their respective appearances were dependent on the sample gas such as synthetic air, nitrogen, or argon."

**Line 277 & Fig. 3: How high was the tropopause? Could be nice to indicate the range of its heights in Fig. 3.**

The tropopause was typically between 10 and 11 km. We added the tropopause range to Fig. 3 and adapted the caption accordingly:

p. 12, l. 304f.: "The grey shadow indicates the range of the tropopause."

We also added this information in the opening of Sect. 3.1.2:

p. 12, l. 286ff.: "Figure 3 shows the altitude dependence of the most abundant ions and of gaseous sulfuric acid. Furthermore, the range of the tropopause (10 to 11 km) is indicated by a grey shadow."

Fig. 3: Not sure of "ncps" is the most intuitive unit for the "normalized count rate" in this case. As count rates are simply divided by the total count rate, the result is just a (unitless) fraction, as indicated also in the main text actually. (Of course, one could in principle multiply with "1 cps" ... but I don't see a conceptual reason for that.)

For more clarification, we added the mention of the normalised count rates in Sect. 2.3:

p. 8, l. 187f.: "The integrated peak values were normalised by the total ion count, yielding the normalised count rates, *nCR*."

Furthermore, we changed the labels of the x-axes in Fig. 3 (a) to (c) to nCR(respective ion) and adapted the caption accordingly:

p. 12, l. 301ff.: "Figure 3: Altitude-resolved box plots of the normalised count rates *nCR* of (a)  $NO_3^-$ , (b) (HNO3) $NO_3^-$ , and (c) HSO4- (all dimensionless), and (d) the number concentration of gaseous H2SO4 (in cm-3)."

**Technical comments:**

**Line 235: "However" appears not to fit.**

We changed it to "Moreover".

**Line 269: "comparably" probably the wrong word.**

We changed it to "comparatively".

Line 302: "points between the 25 % and the 75 % percentiles, i.e. the boxes contain the medium half of the data points" ... That's all fine. But FYI, I believe that thing can also be called the "interquartile range" (or IQR). Not sure how prevalent the term is in our field.

We changed it to "interquartile range" (also for Fig. 5).

**Line 285: See above (L269) ... "relatively" or "comparatively"?**

We changed it to "comparatively".

**References**

- Arnold, F. and Qiu, S.: Upper stratosphere negative ion composition measurements and inferred trace gas abundances, Planetary and Space Science, 32, 169–177, https://doi.org/10.1016/0032-0633(84)90151-X, 1984.
- Arnold, F. and Fabian, R.: First measurements of gas phase sulphuric acid in the stratosphere, Nature, 283, 55–57, https://doi.org/10.1038/283055a0, 1980.
- Beck, L. J., Schobesberger, S., Sipilä, M., Kerminen, V.-M., and Kulmala, M.: Estimation of sulfuric acid concentration using ambient ion composition and concentration data obtained by atmospheric pressure interface time-of-flight ion mass spectrometer, Atmospheric Measurement Techniques, 15, 1957–1965, https://doi.org/10.5194/amt-15-1957-2022, 2022a.
- Beck, L. J., Schobesberger, S., Junninen, H., Lampilahti, J., Manninen, A., Dada, L., Leino, K., He, X.-C., Pullinen, I., Quéléver, L., Franck, A., Poutanen, P., Wimmer, D., Korhonen, F., Sipilä, M., Ehn, M., Worsnop, D., Kerminen, V.-M., Petäjä, T., Kulmala, M., and Duplissy, J.: Diurnal evolution of negative atmospheric ions above the boreal forest: From ground level to the free troposphere, Atmospheric Chemistry and Physics, 22, 8547–8577, https://doi.org/10.5194/acp-2021-994, 2022b.
- Dörich, R., Eger, P., Lelieveld, J., and Crowley, J. N.: Iodide CIMS and m/z 62: the detection of HNO3 as NO3- in the presence of PAN, peroxyacetic acid and ozone, Atmos. Meas. Tech., 14, 5319–5332, https://doi.org/10.5194/amt-14-5319-2021, 2021.
- Gormley, P. G. and Kennedy, M.: Diffusion from a Stream Flowing through a Cylindrical Tube, Proceedings of the Royal Irish Academy. Section A: Mathematical and Physical Science, 52, 163–169, available at: https://www.jstor.org/stable/20488498, 1949.
- Heinritzi, M., Simon, M., Steiner, G., Wagner, A. C., Kürten, A., Hansel, A., and Curtius, J.: Characterization of the mass-dependent transmission efficiency of a CIMS, Atmos. Meas. Tech., 9, 1449–1460, https://doi.org/10.5194/amt-9-1449-2016, 2016.
- Hoell, J. M., Davis, D. D., Jacob, D. J., Rodgers, M. O., Newell, R. E., Fuelberg, H. E., McNeal, R. J., Raper, J. L., and Bendura, R. J.: Pacific Exploratory Mission in the tropical Pacific: PEM-Tropics A, August-September 1996, J. Geophys. Res., 104, 5567–5583, https://doi.org/10.1029/1998JD100074, 1999.
- Hoell, J. M., Davis, D. D., Liu, S. C., Newell, R. E., Akimoto, H., McNeal, R. J., and Bendura, R. J.: The Pacific Exploratory Mission-West Phase B: February-March, 1994, J. Geophys. Res., 102, 28223–28239, https://doi.org/10.1029/97JD02581, 1997.
- Junninen, H., Ehn, M., Petäjä, T., Luosujärvi, L., Kotiaho, T., Kostiainen, R., Rohner, U., Gonin, M., Fuhrer, K., Kulmala, M., and Worsnop, D. R.: A high-resolution mass spectrometer to measure atmospheric ion composition, Atmos. Meas. Tech., 3, 1039– 1053, https://doi.org/10.5194/amt-3-1039-2010, 2010.

- Kürten, A., Rondo, L., Ehrhart, S., and Curtius, J.: Performance of a corona ion source for measurement of sulfuric acid by chemical ionization mass spectrometry, Atmos. Meas. Tech., 4, 437–443, https://doi.org/10.5194/amt-4-437-2011, 2011.
- Lee, B. H., Lopez-Hilfiker, F. D., Veres, P. R., McDuffie, E. E., Fibiger, D. L., Sparks, T. L., Ebben, C. J., Green, J. R., Schroder, J. C., Campuzano-Jost, P., Iyer, S., D'Ambro, E. L., Schobesberger, S., Brown, S. S., Wooldridge, P. J., Cohen, R. C., Fiddler, M. N., Bililign, S., Jimenez, J. L., Kurtén, T., Weinheimer, A. J., Jaegle, L., and Thornton, J. A.: Flight Deployment of a High-Resolution Time-of-Flight Chemical Ionization Mass Spectrometer: Observations of Reactive Halogen and Nitrogen Oxide Species, J. Geophys. Res. Atmos., 105, 3527, https://doi.org/10.1029/2017JD028082, 2018.

**Referee 2**

(Comments by the referee are in **bold font**, answers by the authors are in regular font)

The manuscript entitled "Mass spectrometric measurements of ambient ions and estimation of gaseous sulfuric acid in the free troposphere and lowermost stratosphere during the CAFE-EU/BLUESKY campaign" by Zauner-Wieczorek and co-authors presents ion measurements in the UTLS region. To this end, an API-ToF-MS was operated onboard the HALO aircraft and sampled air masses primarily over Western Europe. The negative ion mode was found to be dominated by NO3- and HSO4- as well as clusters thereof. Based on the measured ion concentrations the number concentration of sulfuric acid was derived. The positive ion mode was studied in less detail but protonated pyridine was identified as a major ion. Based on the data presented an increase of nitrate ions with altitude was found while hydrogen sulfate ions as well as sulfuric acid showed a more evenly distributed trend. My overall assessment of this manuscript is quite positive and it clearly fits the scope of ACP. Especially the introduction is well written and gives a nice overview of previous work, however, a few things need clarification and improvement before final approval.

We would like to thank the Referee for their valuable feedback. We believe that, thanks to the Referee's suggestions, the manuscript could be improved significantly.

Let me start with section 3.1.3 which I feel least comfortable with. While the in-cloud measurement shows some interesting features, the interpretation seems speculative and immature to me. In a way the section sounds vague and does not quite fit the rest of the manuscript. Apart from the fact that this was a one-time signal over 30 seconds only, there are a couple of questions that need clarification.

We agree with the Referee's view that section 3.1.3 holds the least firm results of our manuscript. We are, nevertheless, convinced that also observations that cannot be explained in full detail yet should be communicated to the scientific community to raise awareness of open questions and stimulate future research. By using terms like "we speculate that" we clearly mark our attempt to explain the observations as hypotheses that are still to be debated. To emphasise this, we also added this opening sentence to the section:

p. 14, l. 353: "In this section, we report an interesting finding that may be attributed to an artefact."

As I understand this measurement took place at an altitude of >5km. I actually can't believe that outside temperature at this height and latitude will be positive. What does the temperature reading refer to? What were temperatures during other flights/altitudes?

The reported temperature is the ambient temperature measured by the BAsic HAlo Measurement And Sensor system (BAHAMAS), which is HALO's standard system to collect basic meteorological and flight data. In the initial manuscript, we had erroneously used the Total Air Temperature, which is the uncorrected temperature measured outside the aircraft. We are thankful to the Referee for pointing out this mistake. In the revised manuscript, we use the Static Air Temperature instead, which is the corrected, "true" ambient temperature. The value now reads 261 K instead of 275 K and is, thus, clearly below the freezing point. We changed Fig. 3, the main text in Sect. 3.1.3, the abstract, and the conclusion accordingly:

p. 2, l. 21ff.: "During the transit through a mixed phase cloud, we observed an event of enhanced ion count rates and aerosol particle concentrations that can largely be assigned to nitrate ions and particles, respectively; this may have been caused by the shattering of liquid cloud droplets on the surface of the aircraft or the inlet."

p. 15, l. 363ff.: "The temperature, however, was constant at 261 K, only decreasing by less than 1 K during the humidity peak events. At this temperature, mixed-phase clouds consist mainly of liquid cloud droplets because the most common ice nucleating particles, consisting of mineral dust, become active at lower temperatures (Hoose et al., 2010; Hoose and Möhler, 2012; Kanji et al., 2017)."

p. 15, l. 373ff.: "Because the aircraft passed through a mainly liquid mixed-phase cloud during this event, it is likely that the shattering of liquid cloud droplets on the surface of the aircraft or the sampling system..."

p. 16, l. 404: "During the transit through a mixed-phase cloud, we observed an event..."

Furthermore, we adapted the values for  $n_+$  and  $t_{rec}$  in the main text that were re-calculated based on the new values for  $\alpha$  (after Eq. (2) and (5)):

p.13, l. 308f.: "The average  $t_{rec} = 136$  s (129 to 151 s) and the average  $n_+ = 4090$  cm-3 (3880 to 4540 cm-3)."

This mistake also influenced the calculated values of [H2SO4] after Eq. (1b) because the parameterised value for the ion-ion recombination coefficient,  $\alpha$ , is dependent on the temperature. We corrected the temperature values and, simultaneously, changed the parameterisation of  $\alpha$  from the one by Brasseur and Chatel (1983) to the one by Israël (1957). This is because, in the meantime, we have found a misprint in Israël's formula, which, if corrected, yields an even more favourable parameterisation than the one by Brasseur and Chatel (Zauner-Wieczorek et al., 2021). However, the corrected values for [H2SO4] do not differ significantly from the previously reported ones. For instance, the average concentration for the altitude bin of 13.4 km is now  $1.9 \cdot 10^5$  cm-3 (instead of  $1.8 \cdot 10^5$  cm-3) and for the altitude bin of 8.7 to 9.2 km, it is now  $7.8 \cdot 10^5$  cm-3 (instead of  $9.1 \cdot 10^5$  cm-3). The conclusions drawn from these results are still the same.

In the revised manuscript, the parameterisation by Israël (1957) is introduced (Sect. 2.4), Fig. 3 is updated for the newly calculated values of  $[H_2SO_4]$  and the values in the main text (Sect. 3.1.2), in the abstract, and in the conclusion are corrected accordingly.

On the other hand, reported values of RH exceeding 130% sound completely unfamiliar to me. Basic literature (e.g. Seinfeld & Pandis) reports supersaturations in convective clouds not exceeding 2%, so I'd expect RH to be clearly below 110%. Even if the reported numbers are correct they must be put into context otherwise readers will get confused.

The relative humidity over water is measured by the Sophisticated Hygrometer for Atmospheric ResearCh (SHARC), which employs a tuneable diode laser (TDL). Indeed, the value for *RH* cannot be used reliably during the passage of clouds. We had, therefore, added the sentence "Within clouds, the measurement of the relative humidity can be influenced by the evaporation of cloud particles, thus, relative humidities exceeding 100 % are possible." (old manuscript, p. 6, l. 152f.) to the initial manuscript. While the absolute values are certainly not correct, the relative increase in *RH* during the event of interest can, nevertheless, be demonstrated well. To put this into context, we added the following remarks to the main text:

p. 15, l. 359ff.: "Between 11:01:57 and 11:02:20 UTC, the relative humidity over water (*RH*) showed three peaks of 132–136 % compared to 114 % before and after this event (see Fig. 4 (b)). Please note that the measurements of the relative humidity are influenced by the evaporation of cloud droplets during in-cloud measurements. Thus, the absolute numbers of *RH* are strongly overestimated here. Nevertheless, one can observe the increased peaks in *RH* during the event relative to the measured *RH* values before and after the event."

**In addition, I'd be surprised that during such a number of flights there was only one period of 30 s in-cloud flight. What makes this cloud different from the others?**

During the measurement campaign, we operated the instrument in the Chemical Ionisation (CI) mode for the majority of time, which is subject to other publications. Especially during periods of constant flight levels, we operated the instrument in the Atmospheric Pressure inlet (APi) mode, which is presented in this manuscript. In the APi measurement periods, we rarely changed the flight altitude or passed clouds. Thus, the in-cloud measurement of flight segment 06.2 is unique to our data set despite the large number of flights. We are looking forward to study this phenomenon in more detail in future measurement campaigns where dedicated vertical profiles and cloud-passing flights in the APi mode may be performed.

**Unless this section is improved considerably I'd recommend putting this topic into supplemental material or keep it for another publication when data are clearer.**

We are convinced that, thanks to the valuable feedback by the Referee, this section could be improved and sheds light on a research topic that invites to future research activities.

Along these lines the introduction of C-TOF-AMS and OPC in the instruments section appears quite unexpectedly as they do not relate to the ion (distribution) measurements. These should better be mentioned together with the in-cloud measurements. We are thankful for this suggestion. Section 3.1.3 is concerned with the results and discussion of the in-cloud measurement, while Section 2.1 is concerned with the description of all instruments whose data are discussed within the manuscript. We believe that the description of the instruments should be placed within the Methods section (2.1) to enable interested readers to quickly find the information they are looking for.

**A few minor issues:**

**Page 8, line 182: "...data were averaged to 30 s". What distance does this period relate to at cruise speed? Again, put numbers into context.**

We added the information accordingly:

p. 8, l. 185ff.: "The uncorrupted data were then averaged to 30 s. For typical groundspeeds of 160 to 240 m s-1, this relates to a covered ground distance of 5 to 7 km for one averaged data point."

**Page 9, line 239: "... the value of q applied here MUST be 90%..." This is quite a strong formulation that should be relaxed, maybe by giving a range.**

We chose a less strong formulation:

p. 10, l. 250: "...therefore, we estimated a value for q of 90 % of the maximum polar value of q (Bazilevskaya et al., 2008) for this data set."

**Page 11, Table 2: the exact mass is only given for one ion. Why not show all exact masses for reference? Or do all measured masses agree exactly with the nominal masses?**

All measured masses agree with the expected masses except for the signal at 95.973 that could be assigned to  $SO_4^-$ . Therefore, we added the exact mass of  $SO_4^-$  in the remarks. The exact masses of all observed signals are given in the first column. For more clarity, we changed the caption of Table 2:

p. 11, l. 275: "Table 2: Observed signals in the negative mass spectra with, their exact mass-to-charge ratio, m/z, and the assigned ions."

Regarding section 3.2 "Positive ions": For me and probably for many other readers it would be interesting to see an averaged mass spec of the positive ions. It is shown for negative ions (Figure 2) but not for positive ones. I would very much appreciate it if such a plot could be added.

We agree that a positive mass spectrum is very interesting. Based on the limited data we have in the positive mode, we, however, refrained from adding such a mass spectrum to the manuscript because we want to avoid the impression of a false confidence. Besides the peak for protonated pyridine, the interpretation of the other peaks in the mass spectrum are not yet resolved with confidence. We are looking forward to future studies where we can focus on the positive APi measurements more strongly.

**References**

- Bazilevskaya, G. A., Usoskin, I. G., Flückiger, E. O., Harrison, R. G., Desorgher, L., Bütikofer, R., Krainev, M. B., Makhmutov, V. S., Stozhkov, Y. I., Svirzhevskaya, A. K., Svirzhevsky, N. S., and Kovaltsov, G. A.: Cosmic Ray Induced Ion Production in the Atmosphere, Space Science Reviews, 137, 149–173, https://doi.org/10.1007/s11214-008-9339-y, 2008.
- Brasseur, G. and Chatel, A.: Modelling of stratospheric ions: a first attempt, Annales Geophysicae, 1, 173–185, available at: https://orfeo.kbr.be/bitstream/handle/internal/6155/Brasseur%281983f%29.pdf?sequence =1&isAllowed=y, 1983.
- Hoose, C. and Möhler, O.: Heterogeneous ice nucleation on atmospheric aerosols: a review of results from laboratory experiments, Atmos. Chem. Phys., 12, 9817–9854, https://doi.org/10.5194/acp-12-9817-2012, 2012.
- Hoose, C., Kristjánsson, J. E., and Burrows, S. M.: How important is biological ice nucleation in clouds on a global scale?, Environ. Res. Lett., 5, 24009, https://doi.org/10.1088/1748-9326/5/2/024009, 2010.
- Israël, H.: Atmosphärische Elektrizität: Teil 1. Grundlagen, Leitfähigkeit, Ionen, Akademische Verlagsgesellschaft Geest & Portig K.G., Leipzig, 1957.
- Kanji, Z. A., Ladino, L. A., Wex, H., Boose, Y., Burkert-Kohn, M., Cziczo, D. J., and Krämer, M.: Overview of Ice Nucleating Particles, Meteorological Monographs, 58, 1.1-1.33, https://doi.org/10.1175/AMSMONOGRAPHS-D-16-0006.1, 2017.
- Zauner-Wieczorek, M., Curtius, J., and Kürten, A.: The ion-ion recombination coefficient α: Comparison of temperature- and pressure-dependent parameterisations for the troposphere and lower stratosphere, Atmospheric Chemistry and Physics, in review, https://doi.org/10.5194/acp-2021-795, 2021.

---

## Author Response (AR2)

Answer to Referee comments

Referee 1, round 2

**Comments by the Referee are written in bold font**, answers by the authors are written in normal font

I would like to thank the authors for carefully considering my comments and providing clear responses and clarifications, and the corresponding revisions.

My only remaining suggestion is to still add some indications of the instrument's performance during the deployment to the manuscript, in particular regarding sensitivity and ion transmission. My respective comments were answered well, and reference is made to an upcoming paper that I expect discusses those issues in depth. But I believe that a short discussion (on the order of 2 sentences) is also warranted in this paper -- if necessary only qualitative and of course it can be specific to the particular deployment.

We added the information in the manuscript as suggested by the Referee according to the answers we have provided previously.

The instrument description in Sect. 2.1 now reads as follows:

p. 5, ll. 135ff.: "[...]. Between the inlet line and the mass spectrometer, two pressure stages regulated by mass flow controllers and critical orifices allowed for a constant pressure of 200 hPa in the ion source (in the APi mode, the ion source can be considered as a pressure-controlled pre-chamber in front of the MS). Our inlet system was designed to minimise wall losses; this is beneficial for both chemical ionisation and the ambient ion mode. Nevertheless, certain losses of ions to the inlet walls or in the pressure stages are unavoidable. It can be assumed, however, that the different ions are affected similarly. [...] The TOF-MS records data at a 1 Hz acquisition frequency in an m/z range from 4 to 1121, with a mass resolution of  $\Delta m/m = 2500$  to 3000 and a mass accuracy of 5 ppm for NO3-. From studies on the mass-dependent transmission of the same mass spectrometer (Heinritzi et al., 2016), using a different corona-induced nitrate CI source (Kürten et al., 2011), we estimate a scaling factor of maximum 2 for the transmission difference between NO3- and HSO4- which is within our overall experimental uncertainty."

Moreover, we added the following sentence to the discussion in Sect. 3.1.1:

p. 12, ll. 290f.: "After the CAFE-EU/BLUESKY campaign, we found that the internal voltage settings of the MS can be further improved in order to detect especially the ion clusters with larger m/z values more efficiently."